# Study on the Effect of Customer Psychological Ownership on Value Co-Creation under Service Ecosystem

**Wen Zhou** [1,*] , **Sitan Li** [2] **and Xiangxixi Meng** [3]

1    School of Economics and Management, Beijing Jiaotong University, Beijing 100044, China
2    Moody College of Communication, University of Texas at Austin, Austin, TX 78705, USA; stanli@utexas.edu
3    Century College, Beijing University of Posts and Telecommunications, Beijing 102101, China;
     meng.xiangxixi@grid-elec.com
*    Correspondence: kathyzh123@126.com

**Abstract:** Recently, scholars have tended to study value co-creation from the perspective of service ecosystems, focusing more on the networked, dynamic, and interactive nature of service ecosystems. They believe that the foundation of value co-creation in the service ecosystem is user experience and deep engagement. The deep experience and interaction in the process of value co-creation led to increased psychological ownership, thus increasing the value of users and companies. This study explores the effect of customer psychological ownership on value co-creation from the perspective of deep experience and interaction. The results show that customer psychological ownership and customer fit have a positive effect on the performance of value co-creation, and companies can enhance value co-creation by increasing customers' sense of belonging and tacit understanding.

**Keywords:** value co-creation; service ecosystem; customer psychological ownership; customer fit

## 1. Introduction

In recent years, the research on value co-creation has tended to be networked and dynamic, emphasizing the interaction and integration of research participants in the service ecosystem. Still, there are insufficient empirical studies on service ecosystem value co-creation. Some scholars believe that the basis of value co-creation is to enhance the customer experience from the perspective of the service ecosystem [1]. Customer psychological ownership and customer fit are a reflection of the deeper customer experience. Ownership brings a sense of belonging to the company, while customer fit expresses the depth of interaction between the company and the customer. It has been found that the most direct credit for value growth from a company's perspective comes from customer loyalty, meaning that customer loyalty can be used as a measure of value co-creation outcomes [2,3]. In view of this, this paper investigates the influence of customer psychological ownership on value co-creation under service ecosystems, with customer fit as a mediating variable and brand loyalty as a measure of the outcome of value co-creation. According to the study, the sense of efficacy, self-identity, and spatial demand in customer psychological ownership has a positive effect on brand loyalty, and customer brand fit is the mediating variable.

## 2. Theoretical Basis

### 2.1. Value Co-Creation Concept

Prahalad and Ramaswamy [4] introduced the concept of value co-creation, where companies can gain a unique competitive advantage through value co-creation (co-creation of value with consumers), and consumers can gain satisfaction and unique experiences.

The current conceptual analysis of value co-creation by scholars is divided into four categories:

(1)    Production value co-creation

Some scholars define the concept of value co-creation in the field of production, and Ramirez [5] proposes the concept of value co-production. Thomke and Von Hippel [6] proposed that customers participate in the production of manufacturing to achieve value co-creation with the firm. Etgar [7] proposes a model of consumer co-production that views co-production as a dynamic process.

(2)    Value co-creation of service-led logic

With the gradual enhancement of the customer's role, the customer is involved in more parts of the process and creates value together with the company. Vargo and Lusch [8] propose service-dominant logic. Grönroos [9] proposed the idea of full-service logic in which companies are involved in the daily consumption of customers.

(3)    The whole process of value interaction between business and consumers

Sheth [10] believed that value co-creation includes the interaction and cooperation between consumers and firms in the value creation process of product or service design, production, and consumption.

(4)    Value co-creation based on Service ecosystem

Vargo and Lusch [8] shifted the study of value co-creation from a service science perspective from a binary relationship to a network relationship, emphasizing interactions within and between service systems. Edvardsson et al. [11] suggested that value creation arises in more complex contexts, and they proposed the Service ecosystem in which participants interact and integrate resources. Vargo and Lusch [12] contend that participants exchange services in a service ecosystem, where participants have different value propositions and interact with each other. From a study of nine brand communities, Schau et al. [13] summarized four types of ways in which value is created in communities, namely social network building, impression management, community volunteering, and brand usage, which derive directly from consumer contributions. Lusch et al. [1] proposed that the basis of value co-creation from the Service ecosystem perspective is to enhance the customer's experience. Hatch and Schultz [14] suggested that brand value co-creation is the result of network relationship interaction and is dynamic.

In general, there is a trend of research on value co-creation of service ecosystems, but there are relatively few in-depth empirical studies from the perspective of customer experience.

### 2.2. Customer Psychological Ownership Concept

Pierce et al. [15] concluded that psychological ownership contains both cognitive and affective components, reflecting cognitive components such as awareness, thoughts, and beliefs about the ownership of the target, along with affective components such as pleasure, efficacy, and competence, whereas legal ownership involves only cognitive components. They define psychological ownership as a state of mind in which an individual considers a target or a part of it as "his or her own". They define customer psychological ownership as a state of mind in which consumers consider a company, brand, product, service, etc., or a part of it as "their own", emphasizing the customer's sense of ownership of the target object related to consumption.

### 2.3. Customer Psychological Ownership and Value Co-Creation

With the interactive and customer experience-based characteristics of value co-creation based on the service ecosystem, Pierce et al. [15] found that psychological ownership leads to organizational citizenship behavior. Lyu [16] argued that a sense of membership and belonging helps bring brands closer to their customers. Baxter et al. [17] suggested that participation in value co-creation increases psychological belongingness. Kou and Powpaka [18] identified that interactive communication between customers and firms could lead to an increase in psychological ownership. Kumar's study [19] shows that psychological ownership with the brand and the brand community has a direct effect on

customer engagement with the brand and the brand community, respectively. A brand-based value-congruity has a direct effect on brand engagement, and community-based value-congruity has an indirect effect on brand community engagement through brand community psychological ownership. As psychological ownership is closely related to citizenship behavior, sense of belonging, and customers' inner experience, psychological ownership is set as the independent variable that affects the results of value co-creation.

### 2.4. Customer Fit Concept

Customer fit mainly stressed the customer's commitment to a company's branded product and the customer's willingness to use the product consistently. Brodie et al. [2] found that customer fit is a psychological characteristic of the process of creating a better customer experience through interaction between the customer and the company. Vivek et al. [20] identified customer fit as expressing the depth of the customer-enterprise interaction.

### 2.5. Customer Fit and Value Co-Creation

Brodie et al. [2] contended that in the framework of service-dominant logic, the customer experience process of value co-creation might produce customer fit. That is, customer fit is important for the outcome of value co-creation.

Stockstrom et al. [21] argued that customer fit could contribute to value co-creation at a micro-level. Hollebeek et al. [22] suggested that customer fit has a significant role in both the enhancement of customer experience and the effectiveness of value co-creation.

### 2.6. Brand Loyalty and Value Co-Creation

Brand loyalty refers to a customer's specific preference for a particular corporate brand, mainly emphasizing personal preference. Newman and Werbel [23] concluded that brand loyalty is an attitude of customers who repeatedly buy and only consider buying products of a particular corporate brand.

Brodie et al. [2] concluded that customer loyalty has a positive impact on firm value. Pansari and Kumar [3] believed that for firms, the most direct credit for value growth comes from customer loyalty, and they argued that customer loyalty could be used as a direct indicator of firm value addition. Cossío-Silva et al. [24] concluded that value co-creation leads to users' attitudinal and behavioral loyalty. Hollebeek [25] considered that brand co-creation with customer involvement is positively related to brand loyalty.

## 3. Research Hypothesis and Model

### 3.1. Customer Psychological Ownership and Customer Fit

From the perspective of the service ecosystem, the basis of value co-creation is to enhance customer experience, and customer psychological ownership and customer fit are the response to the deeper experience of customers. Some scholars use customer psychological ownership as an antecedent variable of customer fit on the relationship between the two. Prentice et al. [26] concluded that identity has a positive effect on customer fit. Verleye et al. [27] suggested that the customer's input as one of the roles can be used as an antecedent variable of customer fit.

Pierce et al. [15] verified that customer psychological ownership is divided into three dimensions: the sense of efficacy, self-identity, and space needs in terms of psychological motivation and pathways. In this paper, the measurement of customer psychological ownership is carried out from these three aspects.

Patterson et al. [28], Hollebeek [25], Brodie et al. [2], and other scholars concluded through numerous studies and concluded that customer fit is divided into three dimensions: cognitive, affective, and behavioral. They considered that consumers' attitudes towards corporate brand products change with these three dimensions. In this paper, the measurement of customer fit is analyzed in accordance with these three dimensions because they take into account both psychological and behavioral aspects.

Based on the above research on customer psychological ownership and customer fit by domestic and foreign scholars, the following hypothesis is proposed:

**Hypothesis 1 (H1).** *Customer psychological ownership has a positive effect on customer fit.*

### 3.2. Customer Fit and Brand Loyalty

Numerous scholars have studied customer fit as an antecedent variable of brand loyalty. So et al. [29], Piligrimienė et al. [30], and Dessart et al. [31] suggested that brand loyalty is the result of the influence of customer fit, especially at the level of increasing firm value. Jaakkola and Alexander [32] concluded that customer fit increases brand loyalty while positively influencing people's word of mouth about the brand. Bowden [33] proposed that customer fit can improve the reputation of a brand.

Oliver [34] found through extensive research that brand loyalty is divided into three dimensions: attitude, behavior, and perception. Therefore, this study uses these three dimensions as research variables to develop a research analysis of brand loyalty.

Based on the above-mentioned research on customer fit and brand loyalty by domestic and foreign scholars, the following hypothesis is proposed:

**Hypothesis 2 (H2).** *Customer fit (cognitive, emotional, and behavioral) has a positive impact on brand loyalty.*

### 3.3. Customer Psychological Ownership and Brand Loyalty

Some scholars have used psychological ownership as an antecedent variable for brand loyalty, and Brodie et al. [2] found that when customer engagement is high, it leads to higher brand loyalty. Karahanna et al. [35] found that psychological ownership of a brand can lead to loyal usage behavior.

Once customers have psychological ownership of a brand, they develop a sense of responsibility and commitment to the brand, actively stand up for the brand, see the brand as an extension of themselves, and have customer citizenship behaviors that greatly increase brand loyalty [36]. Zhang et al. [37] studied customer psychological ownership and brand loyalty but used psychological ownership as a mediator to investigate the effect of customer loyalty on customer engagement behavior. Pino et al. [38] conducted two empirical studies to investigate real peer-to-peer hospitality service experiences and demonstrated that identification with service providers engenders a sense of psychological ownership of the service setting, which in turn enhances customers' attitudinal and behavioral loyalty, and notably this effect occurs only when customers engage in cooperative interactions with their providers.

Based on the above-mentioned research on customer psychological ownership and brand loyalty by domestic and foreign scholars, the following hypothesis is proposed:

**Hypothesis 3 (H3).** *Customer psychological ownership has a positive impact on brand loyalty.*

### 3.4. The Intermediary Role of Customer Fit

Bowden et al. [33] found that the process of customer fit can be seen as a psychological process that enables customers to increase their loyalty to the brand. Through numerous studies, Brodie et al. [2] and Hollebeek [25] concluded that customer fit is a bridge built between customers and corporate brands to communicate with each other through the three measures of cognition, emotion, and behavior.

Li [39], Han, and Yu [40] examined the mediating role of customer fit from the perspective of the online shopping environment and customers, respectively, but not from the perspective of value co-creation, and the dimensions chosen for the study were different. The indicators they chose were more focused on the social dimension.

As can be seen from the above, customer fit mainly refers to the fit between customers and corporate brands, through which customers communicate with each other and invest in cognition, emotion, and action.

Based on the above-mentioned research on customer fit, the following hypothesis is proposed:

**Hypothesis 4 (H4).** *Customer fit (cognitive, emotional, and behavioral) has a mediating role between customer psychological ownership and brand loyalty.*

Based on the above analysis, this study proposes a conceptual model as shown in Figure 1 below:

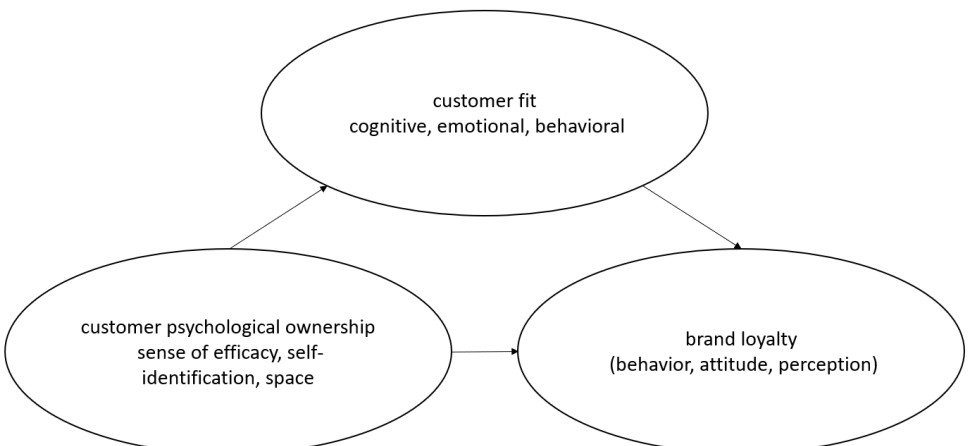

**Figure 1.** Conceptual model.

## 4. Research Methods

### 4.1. Data Collection and Sample

This study mainly analyzes the influence of Gree air conditioner consumers' psychological ownership on the brand loyalty of this air conditioner product under the Service ecosystem. Among them, customer fit is used as a mediating variable for the study. Therefore, a questionnaire was designed in which consumers who had purchased or used Gree air conditioners were the main target of the survey.

The questionnaire for this study was distributed for research on WeChat and Weibo through the WIX app. The questionnaires were filled out by people from all over the country, making the data broader and more realistic. The actual number of questionnaires returned was 268, excluding a total of 14 consumers who had not used Gree air conditioners in the questionnaire. The final valid questionnaire totaled 254, with a valid questionnaire rate of 94.78%.

### 4.2. Variable Measurement

In this study, the relationship between customer psychological ownership and brand loyalty of a Chinese air conditioner company Gree is analyzed, in which customer fit is added as a mediating variable to carry out the corresponding analysis process.

Based on the above, in the process of designing this questionnaire, the design focuses on consumers who have purchased or used Gree air conditioners as the target group for research and analysis. For the questionnaire designed in this study, the main questions of the survey used Likert's five-level scale. The measure of customer psychological ownership is based on the Psychological Ownership Scale designed by Pierce et al. [15]. It contains three dimensions: efficacy (control), self-identity (personal energy engagement), and spatial needs (sense of belonging).

The measurement of customer fit is based on the customer fit scale designed by Brodie et al. [2], and the measure of customer fit contains three dimensions: cognitive, affective, and behavioral.

The measurement of brand loyalty is based on the brand loyalty scale designed by Brakus et al. [41] and the customer loyalty scale developed by Kassim and Abdullah [42], which contains three dimensions: behavioral, attitudinal, and cognitive.

## 5. Data Analysis

### 5.1. Descriptive Statistical Analysis

5.1.1. Descriptive Statistics for the Basic Conditions of the Respondents of the Questionnaire

The questionnaire of this study was designed through the WIX app, and the research was distributed on WeChat, Weibo, and other platforms. People from China filled out the questionnaire to make the data broader, real and convincing. From the results of the descriptive statistical analysis, it can be seen that among the questionnaire research subjects, 52.8% are male, and 47.2% are female. The proportion of males and females is balanced, indicating that both males and females have a certain demand for air conditioners. Since the questionnaires were basically distributed and filled out on WeChat and Weibo platforms, the age of the respondents was relatively young, mainly between 18–25 years old, accounting for 76.4%. From the current length of use of Gree air conditioners, the vast majority are used for more than a year, indicating the popularity of air conditioners and the increasing demand for air conditioners with the development of the social economy. Among them, those who have used it for 1–3 years account for the most, at 44.1%. The concentration of age groups leads to more students in the occupations survey respondents are engaged in, accounting for 56.7%, and the monthly income is correspondingly concentrated below RMB 2000, accounting for 45.3%.

5.1.2. Descriptive Statistical Analysis of Relevant Variables

In this study, each relevant variable was analyzed by questionnaire, and the basic statistical analysis is shown in Table 1 below.

**Table 1.** Basic descriptive statistical analysis of the variables of interest.

| Measurement Indicators | Number | Minimal Value | Maximum Value | Average Value | Standard Deviation | Skewness | | Kurtosis | |
|---|---|---|---|---|---|---|---|---|---|
| | Statistical Quantities | Statistical Quantities | Statistical Quantities | Statistical Quantities | Statistical Quantities | Statistical Quantities | Standard Errors | Statistical Quantities | Standard Errors |
| Aa1 | 254 | 1 | 5 | 4.41 | 0.710 | −1.659 | 0.153 | 5.039 | 0.304 |
| Aa2 | 254 | 1 | 5 | 4.42 | 0.759 | −1.466 | 0.153 | 2.800 | 0.304 |
| Aa3 | 254 | 1 | 5 | 4.45 | 0.708 | −1.513 | 0.153 | 3.600 | 0.304 |
| Ab1 | 254 | 1 | 5 | 4.36 | 0.776 | −1.333 | 0.153 | 2.315 | 0.304 |
| Ab2 | 254 | 1 | 5 | 4.27 | 0.811 | −1.029 | 0.153 | 1.083 | 0.304 |
| Ab3 | 254 | 1 | 5 | 4.33 | 0.811 | −1.437 | 0.153 | 2.703 | 0.304 |
| Ac1 | 254 | 1 | 5 | 4.26 | 0.772 | −0.900 | 0.153 | 0.765 | 0.304 |
| Ac2 | 254 | 1 | 5 | 4.24 | 0.801 | −1.112 | 0.153 | 1.857 | 0.304 |
| Ac3 | 254 | 1 | 5 | 4.09 | 0.986 | −0.873 | 0.153 | 0.034 | 0.304 |
| Ba1 | 254 | 1 | 5 | 4.15 | 0.925 | −0.846 | 0.153 | −0.083 | 0.304 |
| Ba2 | 254 | 1 | 5 | 4.29 | 0.816 | −1.331 | 0.153 | 2.351 | 0.304 |
| Ba3 | 254 | 1 | 5 | 3.89 | 1.036 | −0.521 | 0.153 | −0.652 | 0.304 |
| Bb1 | 254 | 1 | 5 | 4.00 | 1.020 | −0.669 | 0.153 | −0.443 | 0.304 |

**Table 1.** *Cont.*

| Measurement Indicators | Number | Minimal Value | Maximum Value | Average Value | Standard Deviation | Skewness | | Kurtosis | |
|---|---|---|---|---|---|---|---|---|---|
| | Statistical Quantities | Statistical Quantities | Statistical Quantities | Statistical Quantities | Statistical Quantities | Statistical Quantities | Standard Errors | Statistical Quantities | Standard Errors |
| Bb2 | 254 | 1 | 5 | 4.07 | 0.953 | −0.659 | 0.153 | −0.392 | 0.304 |
| Bb3 | 254 | 1 | 5 | 4.03 | 1.015 | −0.695 | 0.153 | −0.416 | 0.304 |
| Bc1 | 254 | 1 | 5 | 4.25 | 0.815 | −1.201 | 0.153 | 1.774 | 0.304 |
| Bc2 | 254 | 1 | 5 | 4.09 | 0.916 | −0.671 | 0.153 | −0.357 | 0.304 |
| Bc3 | 254 | 1 | 5 | 4.20 | 0.870 | −0.992 | 0.153 | 0.696 | 0.304 |
| Ca1 | 254 | 1 | 5 | 4.37 | 0.708 | −0.997 | 0.153 | 1.281 | 0.304 |
| Ca2 | 254 | 1 | 5 | 3.83 | 1.118 | −0.576 | 0.153 | −0.658 | 0.304 |
| Ca3 | 254 | 1 | 5 | 3.81 | 1.148 | −0.690 | 0.153 | −0.479 | 0.304 |
| Cb1 | 254 | 1 | 5 | 4.22 | 0.847 | −0.831 | 0.153 | 0.076 | 0.304 |
| Cb2 | 254 | 1 | 5 | 4.27 | 0.835 | −1.040 | 0.153 | 0.904 | 0.304 |
| Cb3 | 254 | 1 | 5 | 4.17 | 0.946 | −1.031 | 0.153 | 0.648 | 0.304 |
| Cc1 | 254 | 1 | 5 | 4.17 | 0.841 | −0.803 | 0.153 | 0.421 | 0.304 |
| Cc2 | 254 | 1 | 5 | 4.07 | 0.985 | −0.801 | 0.153 | −0.114 | 0.304 |
| Cc3 | 254 | 1 | 5 | 4.28 | 0.821 | −1.067 | 0.153 | 1.110 | 0.304 |
| A | 254 | 1.0000 | 5.0000 | 4.314523 | 0.6288390 | −1.196 | 0.153 | 3.055 | 0.304 |
| B | 254 | 1.0000 | 5.0000 | 4.106737 | 0.7788000 | −0.715 | 0.153 | 0.158 | 0.304 |
| C | 254 | 1.0000 | 5.0000 | 4.131234 | 0.7927743 | −0.724 | 0.153 | 0.088 | 0.304 |
| Valid N (list status) | 254 | - | - | - | - | - | - | - | - |

Note: Aa1–Aa3 represents the sense of efficacy. Ab1–Ab3 represents self-identity. Ac1–Ac3 represents spatial demand. Ba1–Ba3 represents cognitive fit. Bb1–Bb3 represents emotional fit. Bc1–Bc3 represents behavioral fit. Ca1–Ca3 represents behavioral loyalty. Cb1–Cb3 represents attitudinal loyalty. Cc1–Cc3 represents cognitive loyalty. A represents customer psychological ownership. Cc1–Cc3 represents cognitive loyalty. A represents customer psychological ownership. B represents customer to fit. C represents brand loyalty.

From the above table, it can be seen that the mean values of all measures of customer psychological ownership and brand loyalty are above 3.5 and are in the range of 3.5–4.5, which indicates that the research respondents have a high degree of recognition for Gree Air Conditioner. The absolute value of skewness is less than 3, which means that the sample is normally distributed.

5.1.3. Correlation Analysis between Different Variables

The questionnaire of this study is based on the survey related to Gree air conditioner, and the correlation analysis of each variable factor among customer psychological ownership, customer fit, and brand loyalty, in which the Pearson correlation coefficient method is applied to measure the correlation, and the results of the correlation analysis are shown in Table 2 below.

From the correlation analysis among the variable factors in the above table, the correlation coefficients among the variable factors of Gree air conditioner customer psychological ownership, customer fit, and brand loyalty are all positive, which also further indicates that there is a significant positive correlation among the variable factors.



**Table 2.** Correlation analysis between the factors of each variable.

| Factors | Item | Sense of Efficacy | Self-Identification | Sense of Belonging | Cognitive Fit | Emotional Fit | Behavioral Fit | Brand Loyalty |
|---|---|---|---|---|---|---|---|---|
| | | **Relevance** | | | | | | |
| Sense of efficacy | Pearson correlation | 1 | 0.709 ** | 0.631 ** | 0.602 ** | 0.549 ** | 0.659 ** | 0.633 ** |
| | Significance (Bilateral) | - | 0.000 | 0.000 | 0.000 | 0.000 | 0.000 | 0.000 |
| | NNumber | 254 | 254 | 254 | 254 | 254 | 254 | 254 |
| Self-identification | Pearson correlation | 0.709 ** | 1 | 0.742 ** | 0.667 ** | 0.697 ** | 0.726 ** | 0.731 ** |
| | Significance (Bilateral) | 0.000 | - | 0.000 | 0.000 | 0.000 | 0.000 | 0.000 |
| | Number | 254 | 254 | 254 | 254 | 254 | 254 | 254 |
| Sense of belonging | Pearson correlation | 0.631 ** | 0.742 ** | 1 | 0.796 ** | 0.723 ** | 0.744 ** | 0.763 ** |
| | Significance (Bilateral) | 0.000 | 0.000 | - | 0.000 | 0.000 | 0.000 | 0.000 |
| | Number | 254 | 254 | 254 | 254 | 254 | 254 | 254 |
| Cognitive fit | Pearson correlation | 0.602 ** | 0.667 ** | 0.796 ** | 1 | 0.814 ** | 0.765 ** | 0.812 ** |
| | Significance (Bilateral) | 0.000 | 0.000 | 0.000 | - | 0.000 | 0.000 | 0.000 |
| | Number | 254 | 254 | 254 | 254 | 254 | 254 | 254 |
| Emotional fit | Pearson correlation | 0.549 ** | 0.697 ** | 0.723 ** | 0.814 ** | 1 | 0.826 ** | 0.865 ** |
| | Significance (Bilateral) | 0.000 | 0.000 | 0.000 | 0.000 | - | 0.000 | 0.000 |
| | Number | 254 | 254 | 254 | 254 | 254 | 254 | 254 |
| Behavioral fit | Pearson correlation | 0.659 ** | 0.726 ** | 0.744 ** | 0.765 ** | 0.826 ** | 1 | 0.902 ** |
| | Significance (Bilateral) | 0.000 | 0.000 | 0.000 | 0.000 | 0.000 | - | 0.000 |
| | Number | 254 | 254 | 254 | 254 | 254 | 254 | 254 |
| Brand loyalty | Pearson correlation | 0.633 ** | 0.731 ** | 0.763 ** | 0.812 ** | 0.865 ** | 0.902 ** | 1 |
| | Significance (Bilateral) | 0.000 | 0.000 | 0.000 | 0.000 | 0.000 | 0.000 | - |
| | Number | 254 | 254 | 254 | 254 | 254 | 254 | 254 |

** Significantly correlated at the 0.01 level (bilaterally).

*5.2. Reliability and Validity Tests*

5.2.1. Reliability Analysis

The questionnaire was first analyzed for reliability. The Cronbach's alpha value of 0.976 with a > 0.7 was obtained, which passed the test, indicating that the overall data of the questionnaire is reliable and has a very high level of reliability.

Next, reliability test analysis was conducted for each variable factor appearing in the questionnaire. The reliability of customer psychological ownership, customer fit, and brand loyalty were analyzed and the Cronbach's alpha values obtained were 0.924, 0.946, and 0.954, respectively, with a > 0.7, and all reliability was reliable.

5.2.2. Validity Analysis

According to the degree of differentiation, validity analysis is classified as structural validity analysis in quantitative tests. Therefore, the analysis method used is exploratory factor analysis.

Before factor analysis, KMO and Bartlett's sphere tests were performed. The KMO values for customer psychological ownership, customer fit, and brand loyalty were 0.724, 0.755, and 0.767, respectively, which were all greater than 0.7 and suitable for factor analysis.

The contribution of each principal component to the variance of customer psychological ownership, customer fit, and brand loyalty was 79.641%, 86.787%, and 89.368%, respectively, by the principal component analysis method, which met the specified requirements.

From Table 3, it can be seen that the factor loadings of each of the three principal components exceeded 0.5 by the principal component analysis. Among them, self-identification had the largest contribution to the principal component customer psychological ownership, which was 0.919. Sense of efficacy has the smallest contribution to the principal component customer psychological ownership, which is 0.871.

**Table 3.** Component matrix of customer psychological ownership.

| Component Matrix [a] | | |
|---|---|---|
| **Explained Variables** | **Explanatory Variables** | **Ingredients** |
| | | **1** |
| customer psychological ownership | Sense of efficacy | 0.871 |
| | Self-identification | 0.919 |
| | Sense of belonging | 0.887 |
| Extraction method: Principal component analysis. | | |

[a] One component has been extracted.

From Table 4, it can be seen that the factor loadings of each of the three principal components exceeded 0.5 by the principal component analysis method. Among them, emotional fit had the largest contribution to the principal component customer fit, which was 0.946. The cognitive fit had the smallest contribution to the principal component customer fit, which was 0.922.

**Table 4.** Component matrix of customer fit.

| Ingredient Matrix [a] | | |
|---|---|---|
| **Explained Variables** | **Explanatory Variables** | **Ingredients** |
| | | **1** |
| customer fit | Cognitive fit | 0.922 |
| | Emotional fit | 0.946 |
| | Behavioral fit | 0.927 |
| Extraction method: Principal component analysis. | | |

[a] One component has been extracted.

From Table 5, it can be seen that the factor loading of each of the three principal components exceeds 0.5 by the principal component analysis method. Among them, attitude loyalty has the largest contribution to the principal component of brand loyalty at 0.953. Behavioral loyalty has the smallest contribution to the principal component brand loyalty, at 0.935.

**Table 5.** Component matrix of brand loyalty.

| Ingredient Matrix [a] | | |
|---|---|---|
| | | **Ingredients** |
| **Explained Variables** | **Explanatory Variables** | **1** |
| | Behavioral Loyalty | 0.935 |
| Brand loyalty | Attitudinal Loyalty | 0.953 |
| | Cognitive Loyalty | 0.948 |
| Extraction method: Principal component analysis. | | |

[a] One component has been extracted.

The data results obtained from the above analysis of the factors of customer psychological ownership, customer fit, and brand loyalty showed that the scale of this study passed the validity test analysis and met the required requirements.

*5.3. Hypothesis Testing*

5.3.1. Regression Analysis

In this study, based on the results obtained from the factor analysis, the regression equation is presented for the effect of the independent variable of customer psychological ownership and all its related dimensions on the dependent variable of brand loyalty, explaining the relationship between several variables and then conducting the hypothesis test after the next regression analysis.

Firstly, the multiple regression analysis of the three dimensions of customer psychological ownership on brand loyalty.

The influence of three dimensions of customer psychological ownership on brand loyalty was analyzed with the three dimensions of customer psychological ownership, sense of efficacy, self-identification, and sense of belonging as the independent variables and brand loyalty as the dependent variable. The *F*-value obtained from the multiple regression analysis was 155.591. The degree of explanation of customer psychological ownership on brand loyalty was 64.7%, with a significant value of 0.000, which reached a significant level, indicating a significant linear relationship between the independent and dependent variables.

From Table 6, the Sense of efficacy *t*-value obtained from multiple regression analysis was 2.466, Self-identification *t*-value was 4.682, and Sense of belonging *t*-value was 8.050, all with *p*-values less than 0.05, indicating a significant relationship between the independent and dependent variables.

**Table 6.** Regression coefficients of the three dimensions of customer psychological ownership on brand loyalty.

| Models | | Non-Standardized Coefficient | | Standard Coefficient | *t*-Value | Significance |
|---|---|---|---|---|---|---|
| | | **Beta** | **Standard Error** | **Trial Version** | | |
| | (Constants) | −0.048 | 0.211 | - | −0.225 | 0.822 |
| | Sense of efficacy | 0.162 | 0.066 | 0.134 | 2.466 | 0.014 |
| 1 | Self-identification | 0.332 | 0.071 | 0.294 | 4.682 | 0.000 |
| | Sense of belonging | 0.483 | 0.060 | 0.460 | 8.050 | 0.000 |

Note: Dependent variable: brand loyalty.

Next, a multiple stepwise regression analysis of customer psychological ownership, customer fit, and brand loyalty will be conducted, and the model will be validated.

(1) Multiple stepwise regression analysis of customer fit by three dimensions of customer psychological ownership:

The three dimensions of customer psychological ownership, sense of belonging, self-identification, and sense of efficacy, were used as independent variables, and customer fit was used as the dependent variable for multiple stepwise regression analysis. The F-value obtained from the analysis was 200.826, and the explanatory degree of customer psychological ownership on customer fit was 70.3%. The significant values were all 0.000, which reached the significant level. It indicates that there is a significant linear relationship between customer psychological ownership and customer fit.

The analysis yielded a sense of belonging $t$-value of 10.222, self-identification t-value of 4.620, and sense of efficacy $t$-value of 2.346, all with $p$-values less than 0.05. This indicates that there is a significant relationship between customer psychological ownership and there is a significant relationship between customer psychological ownership and customer fit. Therefore, it can be verified that hypotheses H1.1–H1.9 hold.

The regression equation is obtained:

customer fit = 0.536 × Sense of belonging + 0.266 × Self-identification + 0.117 × Sense of efficacy + d

Note: In the regression equation, d represents the constant term.

(2) Multiple stepwise regression analysis of three dimensions of customer fit on brand loyalty:

The three dimensions of customer fit, cognitive fit, emotional fit, and behavioral fit, were used as independent variables. Brand loyalty was used as the dependent variable for multiple stepwise regression analysis. The F-value obtained from the analysis was 551.201, and the explanatory degree of customer fit on brand loyalty was 86.7%. The significant values were all 0.000, which reached a significant level, indicating that there is a significant linear relationship between customer fit and brand loyalty.

The analysis yielded a behavioral fit t-value of 12.764, emotional fit $t$-value of 5.945, and cognitive fit $t$-value of 4.144, all with $p$-values less than 0.05, indicating that there is a significant and significant relationship between customer fit and brand loyalty. Therefore, hypotheses H2.1–H2.3 can be verified to be valid.

The regression equation is obtained:

brand loyalty = 0.541 × behavioral fit + 0.280 × emotional fit + 0.170 × cognitive fit + d

Note: In the regression equation, d represents the constant term.

(3) A multivariate stepwise regression analysis of three dimensions of customer psychological ownership on brand loyalty

The three dimensions of customer psychological ownership, sense of belonging, self-identification, and sense of efficacy were used as independent variables. Brand loyalty was used as the dependent variable for multiple stepwise regression analysis. The F-value obtained from the analysis was 155.591, and the explanatory degree of customer psychological ownership on brand loyalty was 64.7%. The significant values were all 0.000, which reached the significant level. It indicates that there is a significant linear relationship between customer psychological ownership and brand loyalty.

The analysis yielded a sense of belonging $t$-value of 8.050, self-identification $t$-value of 4.682, and sense of efficacy $t$-value of 2.466, all with $p$-values less than 0.05. This indicates that there is a significant relationship between customer psychological ownership and brand loyalty. Therefore, hypotheses H3.1–H3.3 can be verified to be valid.

The regression equation is obtained:

brand loyalty = 0.460 × sense of belonging + 0.294 × self-identification + 0.134 × sense of efficacy + d

Note: In the regression equation, d represents the constant term.

(4) A test of the mediating effect of customer fit on the relationship between customer psychological ownership and brand loyalty

Verify that customer fit has a mediating effect between customer psychological ownership and brand loyalty, with the following steps:

First, it can be seen through the above that the explanation of customer psychological ownership on customer fit is 70.3%, and the significant values are all 0.000, reaching a significant level, indicating that there is a significant linear relationship. It also shows that the independent variable customer psychological ownership in this study can significantly explain the mediating variable customer fit.

Secondly, it is evident from the above that customer fit explains 86.7% of brand loyalty with significant values of 0.000 all reaching a significant level, indicating that there is a significant linear relationship between customer fit and brand loyalty. It also shows that the mediating variable in this study, customer fit, can significantly explain the dependent variable brand loyalty.

Finally, to test the mediating effect of customer fit on the relationship between customer psychological ownership and brand loyalty, the three variables in this study, customer psychological ownership, customer fit, and brand loyalty, were subjected to multiple stepwise regression analyses.

From the Tables 7 and 8, the *F*-value obtained from the analysis is 738.717, and the explanatory degree of customer psychological ownership, customer fit on brand loyalty is 85.4%. The significant values are all 0.000, reaching the significant level. It means that the correlation between customer fit, customer psychological ownership, and brand loyalty are all significant. In the regression model of customer psychological ownership and customer fit on brand loyalty, the coefficient of customer fit is significant, which further indicates the positive effect of customer fit as a mediating variable in the study of the influence of customer psychological ownership on brand loyalty.

**Table 7.** Summary of models for customer psychological ownership, customer fit, and brand loyalty.

| | Model Summary | | | |
| --- | --- | --- | --- | --- |
| Models | R | R Square | Adjustment of R-Square | Standard Error in Estimation |
| 1 | 0.798 [a] | 0.636 | 0.635 | 0.47901 |
| 2 | 0.925 [b] | 0.855 | 0.854 | 0.30331 |

[a] Predicted variables: (Constant), customer psychological ownership; [b] Predicted variables: (Constant), customer psychological ownership, customer fit.

**Table 8.** ANOVA results of customer psychological ownership, customer fit, and brand loyalty.

| | Anova [c] | | | | | |
| --- | --- | --- | --- | --- | --- | --- |
| Models | | Square Sum | Degree of Freedom | Mean Square | *F*-Value | Significance |
| 1 | Regression | 101.187 | 1 | 101.187 | 440.995 | 0.000 [a] |
| | Residuals | 57.822 | 252 | 0.229 | - | - |
| | Total | 159.008 | 253 | - | - | - |
| 2 | Regression | 135.917 | 2 | 67.959 | 738.717 | 0.000 [b] |
| | Residuals | 23.091 | 251 | 0.092 | - | - |
| | Total | 159.008 | 253 | - | - | - |

[a] Predicted variables: (Constant), customer psychological ownership; [b] Predicted variables: (Constant), customer psychological ownership, customer fit; [c] Dependent variable: brand loyalty.

In summary, it was verified that customer fit has a mediating effect between customer psychological ownership and brand loyalty.

Therefore, hypotheses H4.1–H4.3 can be verified to hold.

### 5.3.2. Conclusion of the Hypothesis Test

The results of the hypothesis testing analysis are shown in Table 9 below:

**Table 9.** Results of hypothesis test analysis.

| Hypothetical Question Items | Hypothetical Content | Results |
|---|---|---|
| H1.1 | The higher the sense of efficacy (sense of control) of the customer, the stronger the cognitive fit of the customer. | Established |
| H1.2 | The higher the sense of efficacy, the stronger the emotional fit of the customer. | Established |
| H1.3 | The higher the sense of efficacy, the stronger the behavioral fit of the customer. | Established |
| H1.4 | The higher the customer's self-identification, the stronger the customer's cognitive fit. | Established |
| H1.5 | The higher the customer self-identification, the stronger the customer emotional fit. | Established |
| H1.6 | The higher the customer self-identification (personal energy input), the stronger the customer behavioral fit. | Established |
| H1.7 | The higher the customer spatial demand (sense of belonging), the stronger the customer cognitive fit. | Established |
| H1.8 | The higher the sense of belonging, the stronger the emotional fit. | Established |
| H1.9 | The stronger the customer's behavioral fit, the stronger the customer's sense of belonging. | Established |
| H2.1 | Customer cognitive fit is positively correlated with brand loyalty. | Established |
| H2.2 | Customer emotional fit is positively correlated with brand loyalty. | Established |
| H2.3 | Customer behavioral fit is positively correlated with brand loyalty. | Established |
| H3.1 | The higher the customer sense of efficacy, the stronger the brand loyalty. | Established |
| H3.2 | The higher the customer self-identification (personal energy input), the stronger the brand loyalty. | Established |
| H3.3 | The higher the sense of belonging, the stronger the brand loyalty. | Established |
| H4.1 | The mediating effect of customer cognitive fit on the relationship between customer psychological ownership and brand loyalty. | Established |
| H4.2 | Customer emotional fit mediates between customer psychological ownership and brand loyalty. | Established |
| H4.3 | Customer behavioral fit mediates between customer psychological ownership and brand loyalty. | Established |

Using the above multiple analysis methods, it was found that the hypotheses proposed before the data analysis were all valid.

The above data shows that the standardized coefficient of sense of efficacy is lower for both customer fit and brand loyalty compared to the other variables. The sense of efficacy represents the desire for self-control that customers can choose or control when and how they buy a product according to their own situation. This means that the effect of a sense of efficacy on brand loyalty and customer fit is not very significant. It shows that the sense of efficiency, that is, the customer's desire for self-control, has a relatively insignificant impact on brand loyalty and customer fit.

Ultimately, it is also illustrated that the data from the study shows that under the three variables of customer psychological ownership, customer fit, and brand loyalty, except for a sense of efficacy, which has a $p$-value less than 0.05, which represents reaching standard significance, display $p$-values of the remaining dimensions are less than 0.01, which represents very significant.

## 6. Research Conclusions and Insights

### 6.1. Research Conclusions

A growing number of scholars are turning their attention to the study of value co-creation in the Service ecosystem context, which places more emphasis on interaction, networking, and experience. From the perspective of the service ecosystem, the basis of value co-creation is to enhance customer experience, while customer psychological ownership and customer fit reflect the deeper experience of customers, the sense of belonging brought by ownership can make closer the distance between customers and enterprises, and customer fit expresses the depth of interaction between customers and enterprises.

This study has examined the relationship between customer psychological ownership, customer fit, and brand loyalty in value co-creation activities under the service ecosystem, taking a Chinese air conditioner company Gree as an example. Through the study, it was finally verified that sense of efficacy, self-identification, and spatial demand in customer psychological ownership had a positive effect on brand loyalty, and customer brand fit was the mediating variable. In the value interaction with customers, companies should understand customer needs and attract customer attention from the three dimensions of sense of efficacy, self-identification, and spatial demand of customer psychological ownership. With the promotion of the three dimensions of cognitive, emotional, and behavioral aspects of customer fit, brand loyalty will be enhanced so as to consolidate market position and win the competition.

### 6.2. Research Gaps and Future Prospects

This paper studies the influence of customer psychological ownership on value co-creation in the service ecosystem environment, and Gree is selected as the research object. There are few empirical studies on value co-creation under the service ecosystem, and this paper is innovative from the perspective of customer experience. In future studies, it would be more comprehensive to extend the study to examine the interaction of online platforms. This study focuses on two aspects of psychological ownership and customer fit, while there may be other factors related to psychological influence in the interaction of value co-creation, which can be further improved and extended in the future. The results of value co-creation in this paper are measured by brand loyalty, which is a somewhat business-focused metric, although it covers both business and customer aspects. In future studies, it is important to include indicators that focus on the increase in customer value so that the impact of value co-creation can be measured in a comprehensive manner.

**Author Contributions:** Investigation, W.Z. and X.M.; Methodology, X.M.; Resources, W.Z. and X.M.; Software, S.L.; Supervision, W.Z.; Validation, S.L.; Visualization, W.Z. and S.L.; Writing—original draft, W.Z. and X.M.; Writing—review & editing, W.Z. All authors have read and agreed to the published version of the manuscript.

**Funding:** This research received no external funding.

**Institutional Review Board Statement:** Not applicable.

**Informed Consent Statement:** Informed consent was obtained from all subjects involved in the study.

**Data Availability Statement:** Data available on request due to restrictions privacy.

**Conflicts of Interest:** The authors declare no conflict of interest.

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
