# Peer review of "Study on the Effect of Customer Psychological Ownership on Value Co-Creation under Service Ecosystem"

_sustainability, doi:10.3390/su14052660_

Round 1

Reviewer 1 Report

The paper is exciting for a reader. And the research itself may be interesting for practitioners and scientists. 
Researchers can make some improvements to make this research and paper more valuable, even correctly prepared paper.
1) Some track changes in the paper should be removed.
2) Page 4, row 14; 15; 17; 18 - Based on the related research. Should be precisely addressed, which research? Above mentioned, or some other. Same form in all hypotheses.
3) Page 5, 4.1. Data collection and sample. This research was performed in one country but mentioned that it is not written in which country. For such arguments and 'emotional' and 'psychological' elements of the model, it is essential to mention in which country researchers explored Gree/Mitsubishi. And it should be elaborated and written in the field 4.1. also, it is crucial for the whole paper. It is not the same where you explore one brand in line with emotional aspects. 
4) At the beginning of the research design, there are four hypotheses, and in research results, we see some supporting assumptions never mentioned before. It should be more clearly elaborated.

Author Response

Response 1: Track changes are removed.

Response 2: Above mentioned research. Already corrected for same form problems.

Response 3: Corrected and elaborated for the whole paper.

Response 4: As the supporting assumptions are taken as subordinate factors which belong to the four main hypotheses, they have already been made clearer at the beginning of the hypotheses.

Thank you for your great help!

Some additional changes also are given in the article to improve its quality.

Reviewer 2 Report

Abstract:

Recently, scholars tend to study value co-creation from the perspective of service ecosystem, focusing more on the networked, dynamic, and interactive nature of service ecosystems. They believe that the foundation of value co-creation in the service ecosystem is user experience and deep 13
engagement.

Introduction:

In recent years, the research on value co-creation has tended to be networked and dynamic, emphasizing the interaction and integration of research participants in the service ecosystem.

Line 67: Vargo and Lusch [12] contend that participants 66
exchange services in a service ecosystem, where participants have different value propositions and interact with each other. 

Line 76: In general, there is a trend of research on value co-creation of service ecosystem, but there are relatively few in-depth empirical studies from the perspective of customer experience.

Line 101:  Rodie [2] found that ...

Line 126: From the perspective of the service ecosystem, the basis of value co-creation is to ...

Research Methods:

Line 229: People from all over "the country" China?

Line 260: ... and brand loyalty, in which the Pearson correlation coefficient method is ..

Line 282: ... values for customer psychological ownership, customer fit, and brand loyalty were ...

Line 286: ... logical ownership, customer fit, and brand loyalty was ...

Line 291: ... Among them, self-identification had the largest contribution to the principal component customer ...

Line 298: ... them, emotional fit had the largest contribution to the principal component customer fit, ...

Line 304: ... loyalty has the largest contribution to the principal component of brand loyalty at ...

Line 305: ... Behavioral loyalty has the smallest contribution to the principal component brand ...

Line 314:  ... ownership and all its related dimensions on the dependent variable of brand loyalty ...

Line 318: ... ownership on brand loyalty ...

Line 319:  ... The influence of three dimensions of customer psychological ownership on Brand ...

Lines 321-322: ... Sense of efficacy, self-identification, and sense of belonging as the independent variables and brand loyalty as the dependent variable ...

Line 324: ... ownership on brand loyalty ...

Lines 328-329: ... From Table 6 above, the sense of efficacy t-value obtained from multiple regression analysis was 2.466, the self-identification t-value was 4.682, and the sense of belonging t-value ...

Lines 332-333: Next, a multiple stepwise regression analysis of customer psychological ownership, customer fit, and brand loyalty will be conducted, and the model will be validated.

Lines 336: The three dimensions of customer psychological ownership, sense of belonging, self-identification, and sense of efficacy, were used as independent variables, and customer fit was used as the dependent variable for a multiple stepwise regression analysis.

Lines 343-344: The analysis yielded a sense of belonging t-value of 10.222, self-identification t-value of 4.620, and sense of efficacy t-value of 2.346, all with p-values less than 0.05.

Lines 354-359: The three dimensions of customer fit, cognitive fit, emotional fit, and behavioral fit, were used as independent variables. Brand loyalty was used as the dependent variable for multiple stepwise regression analysis. The F-value obtained from the analysis was 551.201, and the explanatory degree of customer fit on brand loyalty was 86.7%. The significant values were all 0.000, which reached the significant level, indicating that there is a significant linear relationship between customer fit and brand loyalty.

Lines 360-363: The analysis yielded a behavioral fit t-value of 12.764, emotional fit t-value of 5.945, and cognitive fit t-value of 4.144, all with p-values less than 0.05, indicating that there is a significant and significant relationship between customer fit and brand loyalty. Therefore, hypotheses H2.1-H2.3 can be verified to be valid.

Line 365: brand loyalty=0.541×behavioral fit+0.280×emotional fit+0.170×cognitive fit+d

Lines 369-375: The three dimensions of customer psychological ownership, sense of belonging, self-identification, and sense of efficacy, were used as independent variables. Brand loyalty was used as the dependent variable for multiple stepwise regression analysis. The F-value obtained from the analysis was 155.591, and the explanatory degree of customer psychological ownership on brand loyalty was 64.7%. The significant values were all 0.000, which reached the significant level. It indicates that there is a significant linear relationship between customer psychological ownership and brand loyalty.

Lines 376-379:

The analysis yielded a sense of belonging t-value of 8.050, self-identification t-value of 4.682, and sense of efficacy t-value of 2.466, all with p-values less than 0.05. This indicates that there is a significant relationship between customer psychological ownership and brand loyalty.

Line 381: Brand loyalty = 0.460 × sense of belonging + 0.294 × self-identification + 0.134 × sense of efficacy + d

Line 384-385: A test of the mediating effect of customer fit on the relationship between customer psychological ownership and brand loyalty

Line 386-387: Verify that customer fit has a mediating effect between customer psychological ownership and brand loyalty, with the following steps:

Line Line 393-397: Secondly, it is evident from the above that customer fit explains 86.7% of brand loyalty with significant values of 0.000 all reaching a significant level, indicating that there is a significant linear relationship between customer fit and brand loyalty. It also shows that the mediating variable in this study, customer fit, can significantly explain the dependent variable brand loyalty.

Line 402: Table 7. Summary of models for customer psychological ownership, customer fit, and brand loyalty

Line 404: Table 8. ANOVA results of customer psychological ownership, customer fit, and brand loyalty.

c. Dependent variable: brand loyalty 

Lines 405-411: From the above Table 7 and Table 8, the F-value obtained from the analysis is 738.717, and the explanatory degree of customer psychological ownership, customer fit on brand loyalty is 85.4%. The significant values are all 0.000, reaching the significant level. It means that the correlation between customer fit, customer psychological ownership, and brand loyalty are all significant. In the regression model of customer psychological ownership and customer fit on brand loyalty, the coefficient of customer fit is significant, which further indicates the positive effect of customer fit as a mediating variable in the study of the influence of customer psychological ownership on brand loyalty.

Line 413-414: In summary, it was verified that customer fit has a mediating effect between customer psychological ownership and brand loyalty.

Line 418: H1.1 The higher the sense of efficacy (sense of control) of the customer, the stronger the cognitive fit of the customer. Established

H1.2 The higher the sense of efficacy, the stronger the emotional fit of the
customer. Established

H1.3 The higher the sense of efficacy, the stronger the behavioral fit of the
customer. Established

H1.4 The higher the customer's self-identification, the stronger the
customer's cognitive fit. Established

H1.5 The higher the customer self-identification, the stronger the
customer emotional fit. Established

H1.6 The higher the customer self-identification (personal energy input),
the stronger the customer behavioral fit. Established

H1.7 The higher the customer spatial demand (sense of belonging), the
stronger the customer cognitive fit. Established

H1.8 The higher the Sense of belonging, the stronger the Emotional fit. Established

H1.9 The stronger the customer's behavioral fit, the stronger the
customer's sense of belonging. Established

H2.1 customer cognitive fit is positively correlated with brand loyalty. Established

H2.2 customer emotional fit is positively correlated with Brand loyalty. Established

H2.3 customer behavioral fit is positively correlated with Brand loyalty. Established

H3.1 The higher the customer sense of efficacy, the stronger the Brand
loyalty. Established

H3.2 The higher the customer self-identification (personal energy input),
the stronger the brand loyalty. Established

H3.3 The higher the sense of belonging, the stronger the brand loyalty. Established

H4.1 The mediating effect of customer cognitive fit on the relationship
between customer psychological ownership and brand loyalty. Established

H4.2 Customer emotional fit mediates between customer psychological
ownership and brand loyalty. Established

H4.3 Customer behavioral fit mediates between customer psychological
ownership and brand loyalty. Established

Line 419-432:

Using the above multiple analysis methods, it was found that the hypotheses proposed before the data analysis were all valid. The above data shows that the standardized coefficient of Sense of efficacy is lower for both customer fit and Brand loyalty compared to the other variables. The sense of efficacy represents the desire of self-control that customers can choose or control when and how they buy a product according to their own situation. This means that the effect of sense of efficacy on Brand loyalty and customer fit is not very significant. It shows that the sense of efficiency, that is, the customer's desire for self-control, has a relatively insignificant impact on brand loyalty and customer fit. 
Ultimately, it is also illustrated that the data from the study shows that under the three variables of customer psychological ownership, customer fit, and brand loyalty, except for sense of efficacy, which has a p-value less than 0.05, which represents reaching standard significance, display p-values of the remaining dimensions are less than 0.01, which represents very significant.

Conclusions:

A growing number of scholars are turning their attention to the study of value co-creation in the service ecosystem context, which places more emphasis on interaction, networking, and experience. From the perspective of the service ecosystem, the basis of value co-creation is to enhance customer experience, while customer psychological ownership and customer fit reflect the deeper experience of customers, the sense of belonging brought by ownership can close the distance between customers and enterprises, and customer fit expresses the depth of interaction between customers and enterprises.

This study has examined the relationship between customer psychological ownership, customer fit, and brand loyalty in value co-creation activities under a service ecosystem, taking Gree Air Conditioner as an example. Through the study, it was finally verified that sense of efficacy, self-identification and spatial demand in customer psychological ownership had a positive effect on brand loyalty, and customer brand fit was the mediating variable. In the value interaction with customers, companies should understand customer needs and attract customer attention from the three dimensions of sense of efficacy, self-identification, and spatial demand of customer psychological ownership. With the promotion of the three dimensions of cognitive, emotional, and behavioral aspects of customer fit, brand loyalty will be enhanced so as to consolidate market position and win the competition.

Reference [22]: Newman, J. W.; Werbel, R. A. Multivariate analysis of brand loyalty for major household appliances. J. Mark. Res. 1973, 10, 404-409

Reviewer 3 Report

The paper has good descriptive content and relevant information, critically engaged with literature, material and data.

Even the work is generally well referenced, perhaps the author should introduce some recent references.

Overall, the document is clearly written, well organized, and the authors have tried to approach the subject correctly.

Author Response

Response: Recent references have added.

Thank you for your great help!

Some additional changes also are given in the article to improve its quality.
